# Optimization and Application of CRISPR/Cas9 Genome Editing in a Cosmopolitan Pest, Diamondback Moth

**DOI:** 10.3390/ijms232113042

**Published:** 2022-10-27

**Authors:** Zhen Zhang, Lei Xiong, Chao Xie, Lingling Shen, Xuanhao Chen, Min Ye, Linyang Sun, Xiaozhen Yang, Shuyuan Yao, Zhen Yue, Zhengjiao Liang, Minsheng You, Shijun You

**Affiliations:** 1State Key Laboratory of Ecological Pest Control for Fujian and Taiwan Crops, Institute of Applied Ecology, Fujian Agriculture and Forestry University, Fuzhou 350002, China; 2International Joint Research Laboratory of Ecological Pest Control, Ministry of Education, Fujian Agriculture and Forestry University, Fuzhou 350002, China; 3Ministerial and Provincial Joint Innovation Centre for Safety Production of Cross-Strait Crops, Fujian Agriculture and Forestry University, Fuzhou 350002, China; 4Key Laboratory of Integrated Pest Management for Fujian-Taiwan Crops, Ministry of Agriculture and Rural Affairs, Fuzhou 350002, China; 5Key Laboratory of Green Control of Insect Pests, Fujian Province University, Fuzhou 350002, China; 6BGI-Sanya, BGI-Shenzhen, Sanya 572025, China

**Keywords:** cell lines, *Plutella xylostella*, endogenous promoter, plasmid delivery

## Abstract

The CRISPR/Cas9 system is an efficient tool for reverse genetics validation, and the application of this system in the cell lines provides a new perspective on target gene analysis for the development of biotechnology tools. However, in the cell lines of diamondback moth, *Plutella xylostella*, the integrity of the CRISPR/Cas9 system and the utilization of this cell lines still need to be improved to ensure the application of the system. Here, we stabilize the transfection efficiency of the *P. xylostella* cell lines at different passages at about 60% by trying different transfection reagents and adjusting the transfection method. For Cas9 expression in the CRIPSPR/Cas9 system, we identified a strong endogenous promoter: the 217–2 promoter. The dual-luciferase and EGFP reporter assay demonstrated that it has a driving efficiency close to that of the IE1 promoter. We constructed pB-Cas9-Neo plasmid and pU6-sgRNA plasmid for CRISPR/Cas9 system and subsequent cell screening. The feasibility of the CRISPR/Cas9 system in *P. xylostella* cell lines was verified by knocking out endogenous and exogenous genes. Finally, we generated a transgenic Cas9 cell line of *P. xylostella* that would benefit future exploitation, such as knock-in and multi-threaded editing. Our works provides the validity of the CRISPR/Cas9 system in the *P. xylostella* cell lines and lays the foundation for further genetic and molecular studies on insects, particularly favoring gene function analysis.

## 1. Introduction

Hundreds of immortal insect cell lines have been established since the 1960s for a wide range of applications, including but not limited to recombinant protein expression [1,2,3], screening of agrochemicals [4], as well as studies on insect viromes [5] and endosymbionts [6]. With the rapid development of genome editing technologies, increasing molecular and genetic studies have been performed on insect cell lines, particularly using the CRISPR/Cas9 system [7,8,9].

In the application of the CRISPR/Cas9 system to cell lines, the first thing to consider is the impact of delivery [10]. The two elements (Cas9 and sgRNA) in this system can be expressed after the plasmids enter cells or delivered to cells in the form of ribonucleoproteins (RNP). The way of plasmids expression can rely on lipofection, polymer transfection [11], gene gun [12], or accompanied by baculovirus infection [13]. In the form of RNP delivery, it mainly relies on electroporation [14] and various types of nanomaterials [15,16]. As insect cell lines are primarily used for recombinant protein production, the most widely applied delivery methods are lipofection and baculovirus infection. These two methods can ensure the production of sufficient recombination protein not only because of the excellent delivery efficiency, but also because of strong promoters for protein expression.

Furthermore, the CRISPR/Cas9 system also requires strong promoters to drive Cas9 and sgRNA expression in cell lines for efficient genome editing [17]. The expression of sgRNAs is driven by Pol III-type promoters such as U6 [18] and H1 [19]. Cas9 expression is driven by Pol II-type promoter, such as the IE1 viral promoter. However, due to the immobilization of viral promoter sequences and the effect of epigenetic silencing in stable genetic transformation [20], it makes sense to explore endogenous strong promoters for CRISPR/Cas9 systems. Genes with efficient Pol II promoters have been identified in many insects, including *polyubiquitin* in *Aedes aegypti* [21] and *Tribolium castaneum* [22], *actin 3* in *Bombyx mori* [23], *α1-tubulin* in *Drosophila melanogaster* [24]. These endogenous promoters are often used for high-efficiency expression of recombinant proteins and also be applied to the expression of Cas9 protein with the rise of the CRISPR/Cas9 system [25]. Moreover, in the expression of relevant elements of the CRISPR/Cas9 system, the strong Pol II promoter can also be used for the expression of resistance genes or marker genes, which ensures that the gene-edited monoclonal cells can be filtered for genetic researches by antibiotic treatment [26] or flow cytometry [27].

The diamondback moth (*P. xylostella*), is a lepidopteran pest of the cruciferous family causing huge economic losses every year [28]. The cell lines of *P. xylostella* has been established [29,30]. Previous work on the application of the CRISPR/Cas9 system in *P. xylostella* cell lines showed that it can be used for exogenous gene knockout [18], but with an instable transfection. In addition, the utilization of such a system still faces the lack of strong promoters for the expression of multi-element, e.g., the Cas9 gene, resistance genes (e.g., neomycin, puromycin) and marker genes (e.g., EGFP, DsRed). In this study, we stabilized the transfection efficiency in *P. xylostella* cell lines at different passages at around 60% by trying different transfection reagents, adjusting the transfection method and the number of cells. We also identified a strong endogenous promoter with a driving ability close to IE1 for Cas9 expression. Then, we verified the feasibility of the CRSIRP/Cas9 system based on our constructed pB-Cas9-Neo plasmid and pU6-sgRNA plasmid in *P. xylostella* cell lines. By generating a transgenic Cas9 cell line, this study not only significantly improve the validity of the CRISPR/Cas9 system in *P. xylostella* cell lines, but also develops the *P. xylostella* cell lines into a stable platform that would favors future molecular and genetic studies on insects.

## 2. Results

### 2.1. High Transfection Efficiency of the P. xylostella Cell Lines

Cell state, cell generation, cell number, transfection method, transfection operation and even the concentration, size, and purity of the plasmid used for transfection both will affect the transfection efficiency of cells. Therefore, we first set up the transfection mode with serum and without serum in the liposome transfection (Figure 1A). In the experiments of transfection of pB-EGFP plasmid, which can efficiently express EGFP protein both in vivo and in the *P. xylostella* cell lines [18,31], we tested the transfection efficiency of Lipofectamine™ 3000 transfection reagent (Lip 3000), Cellfectin™ II Reagent (Cellfectin II), FuGENE^®^ HD Transfection Reagent (Fugene) and X-tremeGENE HP DNA Transfection Reagent (Xtreme) in these two ways. It can be seen in the fluorescence images taken under the laser confocal that the fluorescent cells are all brighter, indicating that the driving efficiency of the Hr5/IE1 promoter in the *P. xylostella* cell lines is strong (Figure 1B). After counting the fluorescent cells by the ImageJ software, we found significant differences between the different transfection reagents and between the presence and absence of serum during transfection. The reagents of Xtreme and Fugene have good transfection efficiency, which can reach an average of 47.77% and 52.96% in the serum-free transfection process, and there is a significant difference between them. The transfection efficiency of Cellfectin II and Lip3000 was lower, with an average of 20.94% and 12.09% (Figure 1C). We detected the expression levels of EGFP in different treatments by RT-qPCR, and the data also showed such differences (Figure 1D). Among the four transfection reagents, the presence of serum during transfection significantly affected their transfection efficiency. The transfection efficiencies of Xtreme, Fugene, Cellfectin II and Lip3000 decreased from 52.96% to 36.43%, from 47.73% to 28.87%, from 20.94% to 7.63%, and from 12.09% to 6.91%, respectively. Among them, the transfection efficiency of Cellfectin II dropped significantly during the transfection with serum, even lower than Lip3000, which may be related to the serum-free transfection environment clearly required in the reagent instructions. Correspondingly, such significant difference is also reflected in the results of RT-qPCR (Figure 1D).

We also tested the optimal number of cells during transfection. In the six density groups of 5 × 10^4^, 1 × 10^5^, 1.5 × 10^5^, 2 × 10^5^, 3 × 10^5^, 4 × 10^5^ cells per well (24-well plate), we observed that the cells in the two densities of 5 × 10^4^ and 1 × 10^5^ basically did not grow, and compared with the other densities, these two groups had more cells suspended in the center of 24-well plate, which may be due to factors such as the low cell density, the influence of transfection reagents, and the operation of multiple medium changes during the transfection process, so we did not conduct follow-up tests. The results of the other four treatment groups showed that the best transfection effect was achieved at a cell density of 2 × 10^5^/well (Figure 2A). In addition, screening of cells for transgenic or gene knock-in typically requires more than 10 passages of cells, and the phenomenon of cell aging and apoptosis will aggravate as the generation of cells increasing, which affects the transfection efficiency of cells as well. We therefore examined the effect of cell passage on transfection. It can be seen in the graph of transfection efficiency that the cells of generation 14 can achieve an average transfection efficiency of 62.06% after serum-depleted transfection with Xtreme, which is higher than the 57.70% of generation 35 (Figure 2D). Although there was a significant difference in the expression of EGFP between the cells of generation 14 and the cells of generation 35, there was no significant difference in the transfection efficiency and the observation of fluorescent cells (Figure 2B–D). This shows that the aging of cells at about 20 generations has an influence on the transfection effect, but it does not affect the subsequent cell experiments. This shows that cell aging has an impact on the transfection effect, but the 21-generation culture time does not affect the subsequent cell experiments, which ensures the feasibility of the *P. xylostella* cell lines in the application of the CRISPR/Cas9 system and related experiments that need to be screened.

### 2.2. Identification of a Strong Endogenous Promoters

Based on the above optimized transfection stability and high efficiency, we identified the endogenous promoter in the *P. xylostella* cell lines. Here, we constructed the phylogenetic tree of *HSP70* gene and *polyubiquitin* gene of *P. xylostella* and other insects (Appendix A), a group of potential highly expressed genes of *P. xylostella* were identified based on the distance of the genetic relationship with known highly active genes in other species and the similarity shown by sequence alignment, among them the *Pub* genes were *Px18C00217*, *Px02C00515*, *HSP70* genes were *Px07C00656*, *Px18C00336* and *Px05C00303*. It is worth mentioned that there are multiple splices in the *Px07C00656*, *Px18C00336*, and *Px02C00515* genes, and they are located on the sense strand and antisense strand of the DNA double-stranded. Therefore, we cut out sequences of different lengths in the upstream and downstream of these three genes according to their splices for designing primers. The length of its final amplification and the length within the genome sequence are shown in Appendix A and Figure 3A.

We detected the driving efficiency of different promoters by dual-luciferase reporter assay. The promoter of the control group (P-C) is SV40 promoter. The results showed that among several endogenous promoters, only the promoter of *Px18C00217* (217 promoter) had a high efficiency, but it was significantly lower than that of the Hr5/IE1 promoter and the IE1 promoter (Figure 3B). The promoter of *Px05C00303* (303 promoter) has a higher efficiency than P-C, but significantly lower than that of *Px18C00217*. The activities of the upstream and downstream potential promoters of *Px07C00656*, *Px18C00336* and *Px05C00303* were all low, significantly lower than that of the SV40 promoter (Figure 3B,C). So we selected the promoter of *Px18C00217* gene for truncation analysis, and truncated the promoter to seven fragments of 1462, 1267, 1079, 850, 671, 477, and 263, respectively. Its corresponding amplification primers and amplification results are shown in Appendix A and Figure 3A. The results of dual-luciferase reporter analysis showed that when the promoter was truncated to 1267 bp (217–2 promoter), the driving efficiency of the promoter is the highest, close to the level of Hr5/IE1 promoter (Figure 3D). After the promoter was truncated from 1079 bp to 850 bp, the driving efficiency of the promoter decreased significantly, which indicates that this 229 bp is the key regulatory region of 217 promoter activity.

### 2.3. Characterization of the 217–2 Promoter

We used the online software Genomatix Software Suit (https://www.genomatix.de/, accessed on 10 September 2021) and Alibaba 2.0 databases (http://gene-regulation.com/pub/programs.html, accessed on 7 September 2021) to predict the potential transcription factor binding sites for the promoter sequence. We speculated that C/EBPalP and TBP within the 229 bp may be the key to regulate its driving efficiency (Appendix A).

In addition, we found that the driving efficiency of the Hr5/IE1 promoter is always lower than that of the IE1 promoter in dual-luciferase reporter experiments (Figure 3B,D), we speculated that there may be an antagonistic effect between the Hr5 enhancer and the strong viral promoter (HSV TK promoter) on the *hRluc* (*Renilla reniformis*) luciferase expression vector. So we put the 217–2 promoter, the 303 promoter and the IE1 promoter on a vector to drive the expression of EGFP. We found that in the absence of competition from one type of promoters, the efficiency of the Hr5/IE1 promoter is significantly higher than that of the IE1 promoter and the 217–2 promoter (Figure 4B). The efficiency of the 303 promoter is consistent with the results of dual-luciferase reporter experiment, both of which are significantly lower than the 217–2 promoter (Figure 3B and Figure 4B). It can also be seen in the cytofluorigram that most cells emit weak fluorescence under the driving of the 303 promoter. However, although the promoters of 217–2, Hr5/IE1, and IE1 have significant differences in the driving efficiency, the transfected cells all exist bright green light (Figure 4A), which indicates that the three promoters both have strong driving efficiency in the *P. xylostella* cell lines.

### 2.4. CRISPR/Cas9-Mediated Knockout of Endogenous and Exogenous Gene in P. xylostella Cell Lines

As efficient transfection efficiency was stabilized and strong endogenous promoter was identified. We constructed the pB-Cas9-Neo plasmid for Cas9 expression (Figure 5A). Additionally, based on the pU6-BbsI plasmid, the pU6-gRNA plasmid and pU6-shRNA plasmid were constructed by single-segment and multi-segment enzymatic ligation for precise expression of small RNAs (Figure 5A). This method ensures that the shRNA fragment can be precisely linked to the U6 promoter on a vector, and is easier than Golden Gate cloning. Subsequently, we compared the strength of knockdown of EGFP-targeting RNAi and CRISPR/Cas9 systems. The results show that both two methods can significantly affect the expression of EGFP, and it can be clearly observed that the effect of the CRISPR/Cas9 system is particularly significant (Figure 5B,C), which means that the 217–2 promoter and U6 promoters could drive the expression of Cas9 and sgRNA to knock out EGFP. From the cytofluorigram, the effect of RNAi is not as significant as that of the CRISPR/Cas9 system (Figure 5B), we consider it may be related to the ratio of two plasmids co-transfected. In order to confirm the knock-out effect of endogenous genes by CRISPR/Cas9 system, we selected two targets of sgRNA1 (Ubx) “GGATTCGCCTTACGACGCGT” and sgRNA2 (Ubx) “GGTGGTGGCGAGCAGCAGAA” (Figure 5D) in the *PxUbx* gene of *P. xylostella*. We co-transfected pU6-sgRNA1 (Ubx) plasmid and pU6-sgRNA2 (Ubx) plasmid with pB-Cas9-Neo plasmid, respectively. After one week of screening in Grace’s Insect Cell Culture Medium with 10% FBS containing 10 μg/mL G418 antibiotic, extracted the DNA of cells for the first 477 bp of *PxUbx* sequence amplification. As Cas9 binding to the sgRNA1 (Ubx) and the sgRNA2 (Ubx) to cleavage the sequence of *PxUbx*, indels were generated at the 91 bp and 176 bp where the cleavage site is located. This resulted in the corresponding digestion fragments (385 bp and 91 bp, 301 bp and 176 bp) of *PxUbx* fragments after T7 endonuclease I digestion (Figure 5E). Additionally, in the case of non-targeted sgRNA and no sgRNA, no small fragments are produced after T7 endonuclease I digestion. Therefore, we determined that the CRISPR/Cas9 system based on the pB-Cas9-Neo plasmid and pU6-sgRNA plasmid in the *P. xylostella* cell lines is feasible.

### 2.5. Generation of a Transgenic Cas9 P. xylostella Cell Lines

We further utilized the *piggybac* translocation site on the pB-Cas9-Neo plasmid to generate transgenic Cas9 cell lines. As we co-transfected the pB-Cas9-Neo plasmid and the pHelper plasmid into the *P. xylostella* cell lines, transgenic Cas9 cell lines were obtained after 2 months of screening in Grace’s Insect Cell Culture Medium with 10% FBS containing 10 μg/mL G418 antibiotic. In the transposition fragment of pB-Cas9-Neo plasmid, we designed F and R1 for detection at the start translation site of the Cas9 sequence and the terminator of NLS, and designed R2 outside the transposition fragment (Figure 6A). Then, we extracted the gDNA of transgenic Cas9 cells, pB-Cas9-Neo plasmid transiently transfected cells and normal cells, respectively. Additionally, used primer combinations F+R1 and F+R2 for specific amplification. The results of electrophoresis showed that 5226 bp and 4140 bp fragments could be amplified in the gDNA of pB-Cas9-Neo plasmid transient-transfected cells. There were only the 4140 bp fragment of the Cas9 gene sequence can be amplified from the gDNA of transgenic Cas9 cells (Figure 6B). In normal cells, neither fragment can be amplified. Furthermore, in the results of 2-(4-Amidinophenyl)-6-indolecarbamidine dihydrochloride (DAPI) nuclear staining, it can be seen in the fluorescence images that all the cells fluoresce, which was corresponding to the cells stained with DAPI (Figure 6C,D). These both indicated that the transposable fragment in the pB-Cas9-Neo plasmid has successfully integrated into the chromosome, and there is no false positive situation caused by the residual pB-Cas9-Neo plasmid in cells. Interestingly, at a 100× field of view, we observed that the distribution of EGFP protein is mainly in the cytoplasm, which can be clearly distinguished from the DAPI-stained nuclear (Figure 6C). Additionally, at a 400× field of view, it was found that the cells in non-dividing phase have lower levels of EGFP, while the EGFP content was excessive in dividing cells (larger bodies). Additionally, unlike the equal distribution of nuclei during division, the distribution of EGFP tends to be more concentrated in cells on one side (Figure 6D). In addition, the transgenic Cas9 cells grew more slowly than the normal cells, and the number of suspension cells increased significantly after 2−3 days of culture. We speculated that this may be caused by the uneven distribution of the above-mentioned exogenous proteins affecting the growth state of cells. In conclusion, we successfully screened a transgenic *P. xylostella* Cas9 cell lines, which can reduce the influence of CRISPR/Cas9 system on the transfection and expression of the pB-Cas9-Neo plasmid.

## 3. Discussion

In this study, we optimize the application of CRISPR/Cas9 system on *P. xylostella* cell lines by transfection improvement, identification of strong endogenous promoter, as well as construction of a new Cas9-expressed plasmid for cells screening.

In an effort to apply the CRISPR/Cas9 system in the *P. xylostella* cell lines, we first optimized the effect of lipofection. Lipofection is one of the methods of DNA delivery into cells, which has the advantage of lower cost compared with biolistic transfection [12] and electroporation [14]. Compared to transfection of polymers such as polyethylenimine [11] and calcium phosphate transfection [32], lipofection also has the advantage of being easier to obtain and having less cellular impact. The differences in the lipofection efficiency of different brands make it necessary to filter the best production for a specific cell lines [33]. Therefore, after 4 different transfection reagents have been tested, we found that the transfection efficiency (~60%) of Xtreme in the *P. xylostella* cell lines was the highest. Such a transfection efficiency is slightly inferior to that of HEK 293T cell lines [34]; however, it is significant compared to the transfection efficiency of many other insect cell lines [35,36].

A previously known promoter with high activity in *P. xylostella* cell lines is the IE1 promoter [18,31]. In this study, a strong promoter (the 217–2 promoter) of *polyubiquitin* in *P. xylostella* was identified, with an efficiency close to the IE1 promoter. Although it is inconsistent with the result that the efficiency of this promoter in *S. frugiperda* is significantly higher than that of IE1 [37], the 217–2 promoter is still highly active in the dual-luciferase and EGFP reporter analysis. Such an inconsistency in driving efficiency is believed to be a normal case in different system/organisms. Identification of a strong promoter can not only avoid the decrease in driving efficiency when repeatedly using one type of powerful promoter on plasmids, but also can enable the stable expression of Cas9 by excluding the effect of epigenetic silencing in the subsequent transgenic cells. In addition, the promoter of HSP70 gene (303 promoter) that we searched for in *P. xylostella* by the same method did not show high driving efficiency as expected; it might be an activating promoter that is similar to the same type of promoters found by Chen et al. [38]. This mildly active promoter, however, can also be used as a tool for balanced expression with low fitness cost as described by Samantsidis and co-workers [39].

As a convenient gene editing technology, the CRISPR/Cas9 system has a wide range of application, e.g., gene knockout [40], gene cluster knockout, point editing [41,42], knock-in-based high-throughput screening [43], gene drive [44], etc. In this study, the pB-Cas9-Neo plasmid was constructed for CRISPR-mediated gene editing, which showed a comparable function as reported by Huang et al. [18] to enable the targeted knockdown of EGFP expression. The Neo gene on plasmid was verified to considerably favor cell screening in knocking out the endogenous gene *PxUbx*. The Neo-linked EGFP through T2A can also be applied to screen labelled monoclonal cells. Furthermore, the method of segmented ligation we used in the construction of pU6-shRNA plasmid can also be applied to the plasmid assembly of the tRNA-gRNA expression strategy [45]. Overall, the set of plasmids in this study is believed to facilitate the development and application of the CRISPR/Cas9 system, especially in insects.

To ensure efficient functioning of the CRISPR/Cas9 system, the Cas9 and sgRNA can be co-incubated in vitro and then transported into cells in the form of RNP [46], and the single-unit expression method of the CRISPR/Cas9 system can be applied to insect cell lines as well [47]. However, multiple-plasmid co-transfection is still needed in the case of multi-gene knockout and knock-in. The effect of the CRISPR/Cas9 system may be affected owing to delivery efficiency of different-sized plasmids. The transgenic Cas9 *P. xylostella* cell lines in this study enables gene knock-in and multi-gene targeting in cells more efficient, without delivering the pB-Cas9-Neo plasmid.

Management of cosmopolitan pests such as *P. xylostella* increasingly relies on progresses of molecular evidence. Stable applications of the CRISPR/Cas9 system in cell lines, e.g., generation of transgenic cell lines thus considerably favors to investigate molecular mechanisms underlying various phenotypes of interest by allowing the implementation of diverse editing techniques. In this study, previously unnoticed issues of unstable transfection and lack of strong promoters were addressed, laying the foundation for the application of CRISPR/Cas9 system in *P. xylostella* cell lines. Our work is of great significance for developing the *P. xylostella* cell line into a genetic research platform that not only benefits gene function analysis, but also enrich the knowledge about exploitation of genome editing in cell lines, particularly for insects.

## 4. Materials and Methods

### 4.1. The Subculture and Transfection of the P. xylostella Cell Lines

#### 4.1.1. Cell Subculture

The *P. xylostella* cell line was cultured in a T-25 cell culture flask in constant-temperature cell incubator, at 27 °C, cultured with Grace’s Insect Cell Culture Medium (Invitrogen, Carlsbad, CA, USA) with 10% Fetal Bovine Serum (FBS) (Invitrogen, Carlsbad, CA, USA). When the number of cells in the flask reaches 6−7 × 10^6^ cells, transfer 2 × 10^6^ cells to a new flask after changing the medium.

#### 4.1.2. Cell Transfection

The four transfection reagents we tested were Lipofectamine™ 3000 transfection reagent (Invitrogen, Carlsbad, CA, USA), Cellfectin™ II Reagent (Gibco, Carlsbad, CA, USA), FuGENE^®^ HD Transfection Reagent (Promega, Madison, WI, USA) and X-tremeGENE HP DNA Transfection Reagent (Roche, Basel, Switzerland). During the transfection without serum, the *P. xylostella* cells are transferred to a 24-well plate, and allowed to stand in the plate for one day. Then, the medium in the plate is changed to Grace’s Insect Cell Culture Medium. Next, dilute the plasmid and transfection reagent with Grace medium, mix the plasmid weight (μg) and transfection reagent volume (μL) in a ratio of 1:4. After incubating for 10 min, at room temperature, add 50 μL/well to a 24-well plate. The well plate will be placed in a 27 °C constant-temperature incubator for 24 h, then the medium in the well plate is replaced with Grace’s Insect Cell Culture Medium with 10% FBS, and placed in the incubator again, test after 36 h. The process of transfection with serum is that after the *P. xylostella* cells are transferred into the 24-well plate, without changing the medium, the cells are collected 60 h later for detection after adding the incubation of transfection reagent and plasmid.

In experiments using CRISPR/Cas9 system and RNAi to knock out EGFP, the pB-Cas9-Neo plasmid and pU6 plasmid were added to a 24-well plate at a ratio of 2:3; that is, the dosage of pB-Cas9-Neo plasmid was 200 ng/well, and the dosage of pU6 plasmid was 300 ng/well. In experiments generating transgenic Cas9 cell lines, the pB-Cas9-Neo plasmid and the pHelper plasmid (a transposase plasmid we used before [48]) was added to a 24-well plate in a ratio of 9:1; that is, the dosage of pB-Cas9-Neo plasmid was 450 ng/well, and the dosage of pHelper plasmid was 50 ng/well. The rest of the process is the same as above.

### 4.2. Calculation of Transfection Efficiency of the P. xylostella Cell Lines

We calculated the number of cells in bright field and the number of fluorescent cells by the Cell Counters tool in the Plugins column of ImageJ [49], and divided the former by the latter to obtain the value of transfection efficiency (%). In the serum-depleted and de-serum experiments with different transfection reagents, we used two-Way ANOVA in GraphPad Prism 6.01 to analyze difference between multiple groups (Duncan, *p* < 0.05). In the detection of transfection efficiency of different cell passages, we analyzed the difference by Student’s *t*-test (overall significance level *p* = 0.05) in GraphPad Prism 6.01.

### 4.3. Construction of Phylogenetic Tree and Characterization of Promoters

As a housekeeping gene, ubiquitin is highly conserved in organisms and it was identified that the promoter of gene *Pub* in *Ae. aegypti* has strong driving activity. Therefore, here we choose the *Pub* genes that they have identified in *Ae. aegypti* (GENEBANK accession numbers: XM_021854453. 1, XM_001664215. 2, XM_021854454. 1, XM_021854455. 1, XM_021854452. 1) as alignment genes. Moreover, Chen et al. [38] identified promoters of genes of *HSP70* family with high driving efficiency in *S. frugiperda*, so we selected the *HSP70* genes of *S. frugiperda* (GENEBANK accession numbers are MN735775, MN735776, MN735777, MN735778, MN735779, MN735780) as alignment genes as well. Subsequently, as aligning with the above genes in NCBI database (NCBI BLAST, https://www.ncbi.nlm.nih.gov/, accessed on 23 June 2021) to retrieve related genes and downloading genes according to the annotation of genes in the genome database of *P. xylostella* (DBM-DB, http://iae.fafu.edu.cn/DBM/, accessed on 24 June 2021), we constructed the phylogenetic tree of *Pub* and *HSP70* genes by the MrBayes algorithm in PhyloSuite [50]. The Mafft algorithm in PhyloSuite was used to perform multiple sequence alignment, and through ModelFinder in PhyloSuite, the best models for Bayesian tree building were calculated as: HKY+F+G4 (*Pub*) and SYM+G4 (*HSP70*). Then, the phylogenetic tree is constructed by the MrBayes algorithm in PhyloSuite. The bootstrap was both performed 2000 times.

Potential transcriptional regulatory elements were predicted using the online software Genomatix Software Suit (https://www.genomatix.de/, accessed on 10 September 2021) and Alibaba 2.0 databases (http://gene-regulation.com/pub/programs.html, accessed on 7 September 2021) and annotated in the promoter sequence.

### 4.4. Acquisition of Potential Promoter Regions

The position information of each target gene on the chromosome was queried in *P. xylostella* genome annotation file. The corresponding sequences of these genes in *P. xylostella* genome sequence file was intercepted by the Fasta Subseq (Basic) function in TBtools [51]. The transcription direction of target genes was judged by the alignment of the intercepted sequence and the gene sequence. Then, the upstream 3000 bp opposite to the transcription direction of target genes was intercepted by TBtools as a potential target gene promoter. We used the potential promoter sequence as a template, designed primers by Oligo 7. The Primer efficiency of mismatching in Primer False Priming Sites is controlled within 150, the G value of primer self-matching is within 15, and no hairpin structure is required. Primer synthesis after alignment with no specific binding sites in *P. xylostella* genome database (DBM-DB, http://iae.fafu.edu.cn/DBM/, accessed on 25 July 2021). The final primers used to amplify potential promoters are shown in Appendix A. The gDNA of the *P. xylostella* cells was extracted by genome extraction kit (Tiangen, Beijing, China). The used the *P. xylostella* gDNA as a template, mixed with synthetic primers, and amplified with PrimeSTAR^®^ HS DNA polymerase (Takara, Kusatsu, Japan). The reaction conditions were as follows: 95 °C 3 min; 95 °C 15 s, 55 °C 15 s, 72 °C 2 min, 30 cycles, 72 °C 2 min. The amplification results were verified by agarose gel electrophoresis, followed by sanger sequencing.

### 4.5. Construction of Plasmids

#### 4.5.1. Construction of luc2 (*Photinus pyralis*) Luciferase Expression Plasmids

The *luc2* (*Photinus pyralis*) luciferase vectors containing potential endogenous promoters was constructed by pEASY-Basic Seamless Cloning and Assembly Kit (Transgene, Beijing, China). The process is as follows: plasmids were linearized by inverse PCR amplification, and the sequence 15–20 bp of the end of the linearized vector was added to the 5′ end of the primer that used to amplify the potential promoter fragments. The obtained promoter fragments were used as a template for PCR amplification, and after gelation, it was added to 10 μL of the recombinase action system according to the ratio of linearized vector (pmol): promoter fragment (pmol) to 1:5, 50 °C for 15 min to reconstitute into a plasmid. Final plasmid extraction with Endo-Free Plasmid Mini Kit II (Omega, Madison, WI, USA). The construction method of the EGFP expression vector is the same as above. The above-mentioned primers are shown in Appendix A.

#### 4.5.2. Construction of pB-Cas9-Neo Plasmid

The Neo fragment was from the pCI-Neo plasmid (Youbio, Chongqing, China), and the Cas9 fragment was from the pTD1-T7-Cas9 plasmid. The construction steps were as follows: we took the pB-EGFP plasmid as a template, and we amplified the part except the EGFP expression element. The sequence of T2A was added to the reverse primer of Neo fragment and the forward primer of EGFP fragment for recombination. Then, we amplified the IE1 promoter sequence with homologous arms, and recombined multiple fragments into a pB-Neo-EGFP plasmid. After the sequencing was correct, the plasmid was reversely amplified to linearization, and the 217–2 promoter sequence, Cas9 sequence and SV40 signal sequence with homology arms were amplified, then recombined into pB-Cas9-Neo plasmid, sequencing to ensure the correctness of the plasmid. The primers used for pB-Neo-EGFP and pB-Cas9-Neo plasmid construction are shown in Appendix A.

#### 4.5.3. Construction of pU6-sgRNA Plasmid and pU6-shRNA Plasmid

We added different enzyme cleavage sites (5′-TAGT-3′ and 5′-AAAC-3′) to the sense and antisense strands of the sgRNA fragments, respectively. We mixed the two Oligos with T4 polynucleotide kinase (Vazyme, Nanjing, China), phosphorylated and annealed as per the following procedure: 37 °C 30 min, 95 °C 5 min. The linearized pU6-BbsI was purified after 4 h of digestion with BbsI endonuclease (NEB, UK). The linearized plasmid (pmol) and annealed sgRNA fragments (pmol) were then added to the T4 ligase (Takara, Kusatsu, Japan) reaction at a ratio of 1:10. After 12 h of incubation at 16 °C, transformed into *E. coli* and subsequent bacterial PCR with the sense strand of the sgRNA and the detection primer, sequencing to ensure the correctness of the plasmid. The construction method of pU6-shRNA is similar to that of pU6-sgRNA. The shRNA fragment is divided into two segments for phosphorylation, annealing, and a new ligation site is set at the junction of the two segments, then followed by ligation, transformation, detection and purification in the same way. The related primers used are shown in Appendix A.

### 4.6. Dual-Luciferase Reporter Assay Experiment

The pGL3-Basic plasmid with the endogenous promoter was co-transfected with the pRL-TK plasmid expressing *hRluc* (*Renilla reniformis*) luciferase into the *P. xylostella* cells. On the third day after transfection, we added 100 μL of lysate to each well, lysed for 20 minutes, then collected the lysate, and added 10 μL of lysate per well to a 96-well plate. Luciferase Assay Reagent II from Promega ONE-GloTM Luciferase Assay System (Promega, Madison, WI, USA) was mixed in the 96-well plate in a way of testing 8 wells at a time, and detected the fluorescence intensity with GloMax 96 (Promega, Madison, WI, USA). After all samples were added with Luciferase Assay Reagent II and tested, the Stop & Glo^®^ Reagent was added in the same way and detected fluorescence intensity. We normalized the expression data by dividing the fluorescence intensity value of the first measurement by the magnitude of the second, and performed one-way ANOVA with GraphPad Prism 6.01 to analyze the differences between groups (Duncan, *p* < 0.05).

### 4.7. RT-qPCR Detection

The cells were collected for RT-qPCR detection after the third day following the above-mentioned transfection with serum or without serum procedure. The RNA of the *P. xylostella* cells was extracted by RNA Extraction Kit (Promega, Madison, WI, USA), which was quantitatively reversed to cDNA by FastKing One-Step Reverse Transcription Kit (Tiangen, China). Then, the SYBR Select Master Mix for CFX kit (Omega, Madison, WI, USA) was used for RT-qPCR and detected by CFX96 Touch (Bio-Rad, Hercules, CA, USA). The program was as follows: 50 °C 2 min; 95 °C 2 min; 95 °C 15 s, 60 °C 30 s, 40 cycles. The expression of *RPL32* gene of *P. xylostella* was used as an internal reference [52]. The related primers are shown in Appendix A. In serum-depleted and de-serum transfection experiments, we used two-way ANOVA in GraphPad Prism 6.01 to analyze difference between multiple groups (Duncan, *p* < 0.05). In the detection of different cell passages for transfection, the difference was analyzed by Student’s *t*-test (overall significance level *p* = 0.05). In other experiments, one-way ANOVA was used to analyze difference (Duncan, *p* < 0.05). The analysis methods mentioned above were all carried out in GraphPad Prism 6.01.

### 4.8. DAPI Staining Assay

We transferred 1 × 10^5^ transgenic Cas9 cells to a 48-well plate. After the cells fully adhered, we then changed the medium to 200 μL of DAPI staining solution (Yeasen, Shanghai, China). After incubation at room temperature for 15 min, the cells were washed twice with PBS for fluorescence microscopy observation.

### 4.9. T7 Endonuclease I Digestion

The cells co-transfected with pU6-sgRNA plasmid and pB-Cas9-Neo plasmid were cultured in Grace’s Insect Cell Culture Medium with 10% FBS containing 10 μg/mL G418 antibiotic for 1 week, and the gDNA of it was extracted. We used the gDNA as a template to amplify the fragments around the target. The primers for amplification are shown in Appendix A. The amplified fragments of different treatment groups are purified and quantified at 500 ng and then diluted to 17 μL with pure water. The procedure for hybridization and annealing combined was as follows: 95 °C for 5 min, drop to 85 °C at 1 °C/20 s, drop to 25 °C at 0.1 °C/1 s, and store at 4 °C. Then, we added 1 μL T7 Endonuclease I (Transgene, Beijing, China) and 2 μL T7 Endonuclease I Buffer, incubated at 37 °C for 20 min. We added 2 μL 10× DNA loading Buffer to complete the reaction and run electrophoresis.

## 5. Conclusions

Cell lines serve as a unique platform for gene function analysis. In this study, we improve efficient transfection on the *P. xyostella* cell lines and identify a strong endogenous promoter to ensure the stability of the CRISPR/Cas9 system during delivery and expression. A transgenic Cas9 cell line is also developed to facilitate the exploitation of various gene editing methods. Our work is not only of great significance for developing the *P. xylostella* cell lines into a well-applicable genetic research platform, but also provides insights into the optimization and application of other cell lines, particularly for insect species.

## Figures and Tables

**Figure 1 ijms-23-13042-f001:**
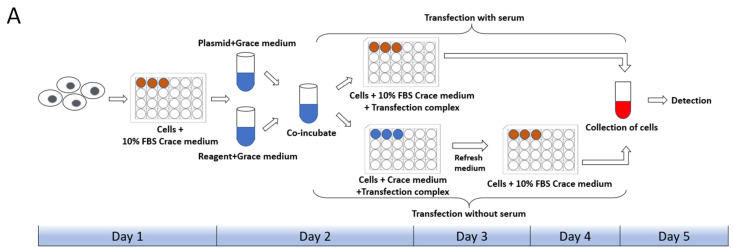
Transfection efficiency of different transfection reagents with serum and without serum. (**A**) Schematic diagram of the workflow with and without serum during transfection. The red in the 24-well plate indicates that the medium contains serum, and the blue indicates that only Grace’s Insect Cell Culture Medium (Grace medium) is present. (**B**) Fluorescence image of cells after transfection. Photographed by laser confocal, exposure time and excitation light intensity are consistent during shooting. (**C**) The relative expression of EGFP under different treatments. The reference gene is *RPL32* of *P. xylostella*, ** indicates highly significant (Duncan, *p* < 0.01). (**D**) The efficiency of cell transfection under different treatments. * indicates significant difference (Duncan, *p* < 0.05), ** indicates highly significant difference (Duncan, *p* < 0.01).

**Figure 2 ijms-23-13042-f002:**
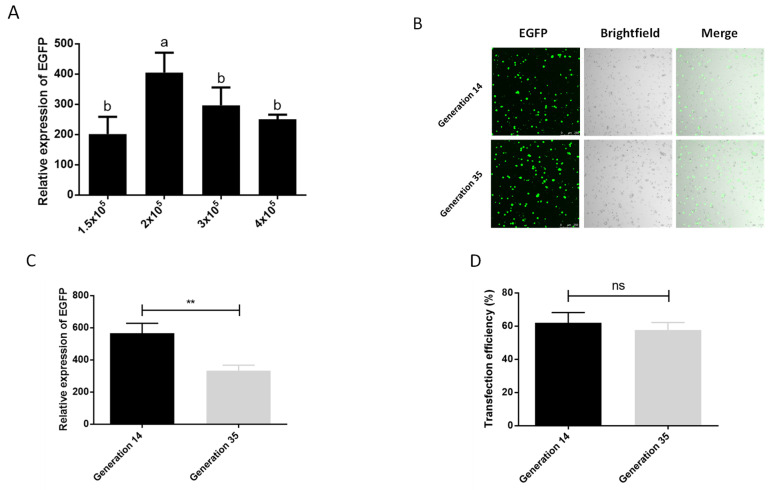
Effect of cell number and passage on transfection efficiency. (**A**) The effect of different cell numbers on the transfection. The significant differences between groups are indicated by lowercase letters (Duncan, *p* < 0.05). (**B**) Fluorescence image of cells after transfection. Photographed by laser confocal, exposure time and excitation light intensity are consistent during shooting. (**C**) The relative expression of EGFP after transfection of cells of different passages. ** indicates highly significant difference (*p* < 0.01). (**D**) Transfection efficiency of cells of different passages. ns means no significant difference.

**Figure 3 ijms-23-13042-f003:**
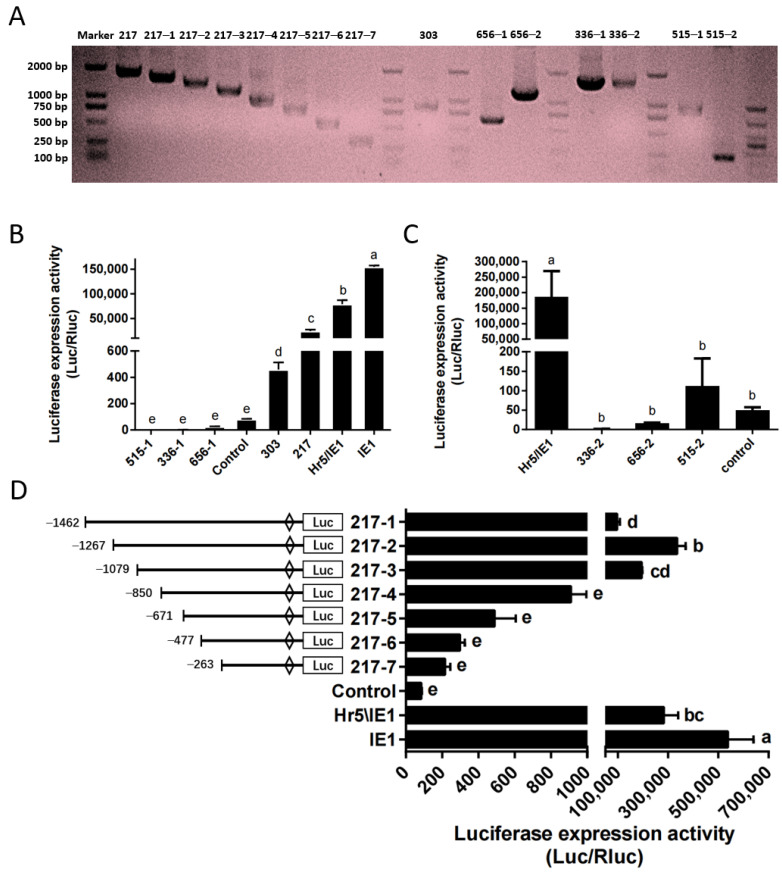
Amplification and efficiency identification of the *P. xylostella* endogenous promoters. (**A**) Amplification of endogenous promoter and truncation of the 217 promoter. (**B**,**C**) Driving efficiency of the *P. xylostella* endogenous promoter, the significant difference between groups are indicated by lowercase letters (Duncan, *p* < 0.05). (**D**) Driving efficiency of the 217 promoter after truncation, the significant difference between groups is indicated by lowercase letters (Duncan, *p* < 0.05).

**Figure 4 ijms-23-13042-f004:**
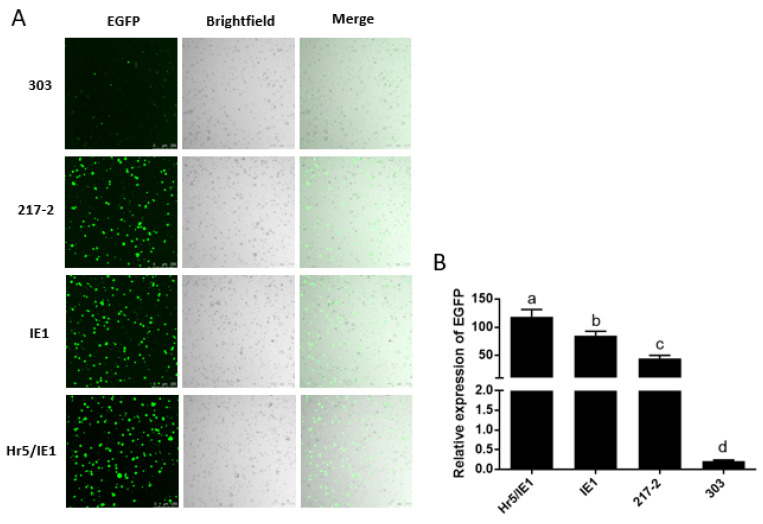
Visualization of the drive efficiency of the 217–2 promoter. (**A**) Fluorescence images of cells with endogenous and exogenous promoters driving EGFP expression. Photographed by laser confocal, exposure time and excitation light intensity are consistent during shooting. (**B**) The relative expression of EGFP driven by different promoters. The significant differences between groups are indicated by lowercase letters (Duncan, *p* < 0.05).

**Figure 5 ijms-23-13042-f005:**
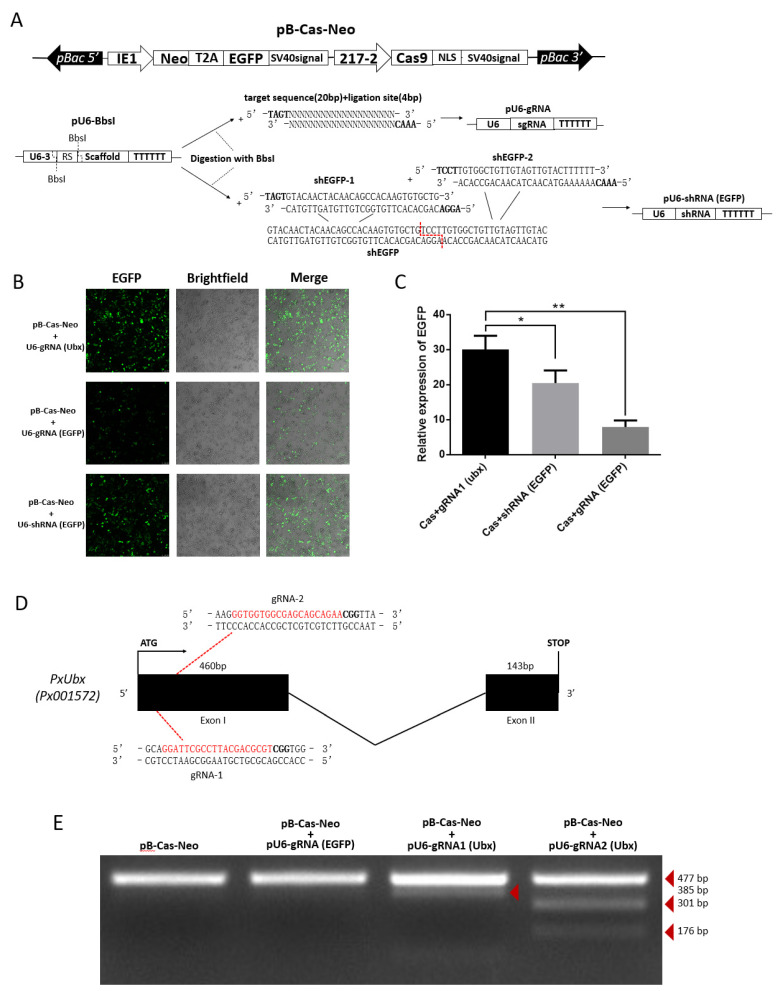
CRISPR/Cas9 system knocks out exogenous and endogenous genes in the *P. xylostella* cell lines. (**A**) Plasmids for RNAi and CRISPR/Cas9 system. (**B**) Fluorescence images of CRISPR/Cas9 system and RNAi knocking out EGFP. photographed by laser confocal, exposure time and excitation light intensity are consistent during shooting. (**C**) The relative expression of EGFP after CRISPR/Cas9 system and RNAi knockout. * indicates significant difference (Duncan, *p* < 0.05), ** indicates highly significant difference (Duncan, *p* < 0.01). (**D**) The *P. xylostella* endogenous gene *PxUbx* and its selected targets. (**E**) Cleavage bands generated by T7 endonuclease I digestion after CRISPR/Cas9 targeting *PxUbx*.

**Figure 6 ijms-23-13042-f006:**
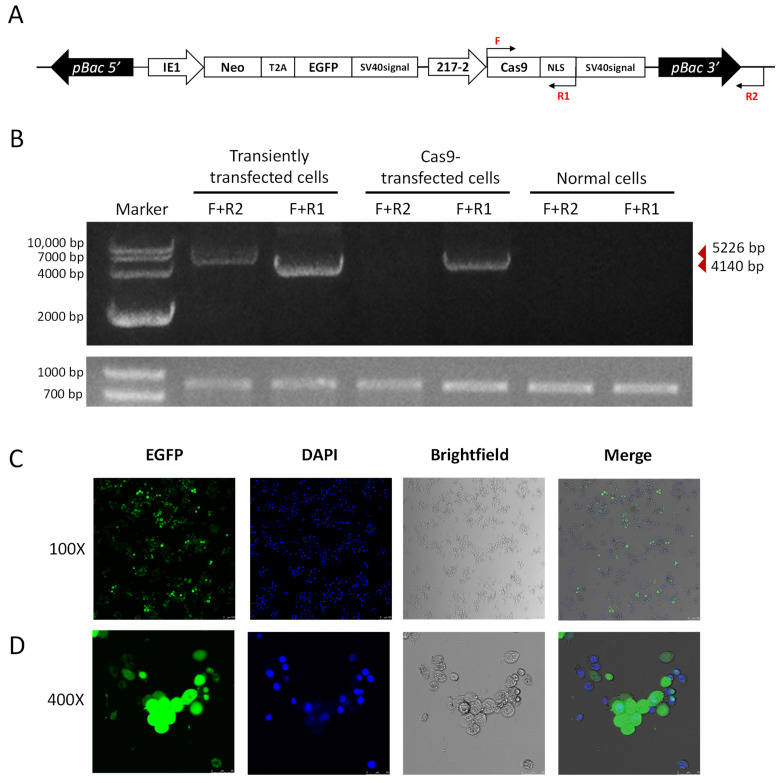
The transgenic Cas9 cell lines of *P. xylostella*. (**A**) The position of the primers for detection in the transgenic Cas9 cells. The amplification direction of F, R and R1 is consistent with the directions of the respective arrows. (**B**) The amplification results of detection primers for transgenic Cas9 and non-transgenic cells. The gel image below is the amplification result of 830 bp in the *Pxubx* gene sequence. (**C**,**D**) Fluorescence images at 100× and 400× after incubation of transgenic Cas9 cells with DAPI staining solution. Photographed by laser confocal, exposure time and excitation light intensity are consistent during shooting.

## Data Availability

All data generated or analyzed during this study are included in this published article (and its Appendix A files).

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
