# Peer review of "Optimization and Application of CRISPR/Cas9 Genome Editing in a Cosmopolitan Pest, Diamondback Moth"

_ijms, 2022, doi:10.3390/ijms232113042_

Round 1

Reviewer 1 Report

The manuscript titled “Optimization and application of CRISPR/Cas9 genome editing in a cosmopolitan pest, diamondback moth by Zhen Zhang , Lei Xiong , Chao Xie , Lingling Shen, Xuanhao Chen , Min Ye , Linyang Sun , Xiaozhen Yang , Shuyuan Yao , Zhen Yue , Zhengjiao Liang , Minsheng You , Shijun You  presents a tool to validate the CRISPR/Cas9 system in P. xylostella cell lines, establishing subsidies for new genetic and molecular studies.

I would like to highlight some observations I make when reading the manuscript.

General aspects: The introduction is well written and presents the subject well. The final objectives and hypotheses of the work can be clarified. The results mix results, methodology and a part of the discussion, this can be tiring for the reader, at the same time that some concepts for understanding the results are missing during the reading. Perhaps the order of the topics should be changed so that this does not happen (for methodology, results and discussion). The discussion is clear and emphasizes the results. The article has merit for publication after going through the revision changes.  In general, the graphics have poor quality, it can be improved.

In Keywords: CRISPR/Cas9 system; This word has already been used in the title.

Line 50 -54; A citation is recommended.

Line 88 - 101; It should be in the Materials and Method section.

Figure 1. The images A and C have poor quality. I suggest the authors to do image A again.

Line 125 - 127; Should be in Materials and Method section.

Line 162-178; It should be in the Materials and Method section.

Present a conclusion at the end of the text.

References; The references support well the ideas presented by the authors. The formatting should be revised.

Reviewer 2 Report

Article number 1912474 submitted to the International Journal of Molecular Sciences, IJMS, MDPI, entitled “Optimization and application of CRISPR/Cas9 genome editing in a cosmopolitan pest, diamondback moth” brought good results. These biotechnological approaches are efficient tools for gene analysis and validation for future genetic and molecular studies in Plutella xylostella. However, the draft lacks the future implications of the study and conclusion. The publication of this study requires corrections, revision and some suggested changes for example are in the comments portion to improve the manuscript. Please find suggested corrections, reference writing, journal-style format, author’s instructions, use of abbreviations and missing information for revision. 

Introduction:

Line 40:  Please replace the word “throughput”

Line 62: Please replace the word “A. aegypti” with  “Aedes aegypti

Line 62-63: Please replace the word, T. castaneum  with Tribolium castaneum

Line 62-63: Please replace the word, B. mori with Bombyx mori

Line 62-63: Please replace the word, D. melanogaster with Drosophila melanogaster

Line 68:  “monoclonal cells can be filtered for genetic researches by antibiotic treatment” [25] Please recheck the reference if this is correctly written

Results:

There are many references in the results portion which need to include in the discussion portion

Line 95-97: The transfected plasmid was pB-EGFP plasmid, which can efficiently express EGFP protein 96 both in vivo and in the P. xylostella cell lines [17, 30].

Line 106-107: Treatments by RT-qPCR. Please use a similar format in writing

Line 114: such differences are shown in RT-QPCR data, Please write a similar format in text

Line 244-246: In the construction method of the shRNA expression plasmid, we optimize the multiple enzyme digestion and ligation method of Huang et al [17]. Rephrase the sentence with clear information for the readers as (multiple enzyme digestion and ligation method) seems confusing

Line 117-123: Fig 1: The word “serum-free” may be replaced by “without serum”

“extremely significant difference” with “highly significant difference”

Line 125-129: “To ensure stable transfection efficiency, we also evaluated the optimal number of cells during transfection. In a 24-well plate, we set up 6 density gradients of 5´104, 1´105, 1.5´105, 2´105, 3´105, 4´105 cells per well for pB-EGFP plasmid transfection. The cells were collected for RT-qPCR detection after the third day following the above-mentioned serum-depleted transfection procedure” Please write these methodologies in the material and methods portion and include only results in this section

Line 136: Please check the word “CRISPR/Cas” if it is correctly written

Figure 2A: Please recheck the lettering of the figure, if this is correct

Line 142, 145: give space in between the words (Fig2.D), (Fig2.B C D).

Line 155: ** indicates highly significant difference

Line 167: [20] Reference, need not give reference as already mentioned

Line 168: Please replace the word A. aegypti with Ae. Aegypti

In supplementary figures please mention which figures are S1 and S2. Please also write the scientific name in italics

Line 203: Figure 3A: Please mention units of 100,250, 500, 750…..

 Please also recheck the units in figure 7B

Units may also be rechecked in all figures of the MS

Line 203: Figure 3B: Please recheck Figure 3B: if the significance letters of 336-2, 656-2 and the control group are correct

Please keep the consistency in writing figure and table numbers as in some places written as Fig.3 and figure 3 A. please also check the spaces in between the words throughout MS

Line 203: Please mention “bp” units in the figure

Line 200-211: Please add citation/reference for the software used (if necessary)

Line 213-217: Figure 4 may be included in the supplementary figures (if necessary/author’s choice)

Line 226, 228, 232: (Fig5. B). please give space in between the words and keep consistency in formatting Figure/ Fig (authors’ instructions/ journal’s style formatting

Line 244-245: Please write the phrase “In the construction method of the shRNA expression plasmid, we optimize the multiple enzyme digestion and ligation method of Huang  et al [17]” in the material and methods portion.

Line 246-248: Please rephrase the sentence “This method ensures that the shRNA fragment ….. precisely link to the U6 promoter on the vector.

Line 259: Please rewrite Grace medium in correct spellings

Figure 6 E: Please mention the “bp” in the figure

Figure 6 E: The significant differences between groups are indicated by lowercase letters.

these letters are not visible in the figure. Please rephrase the sentence as significance is expressed in *

Line 275: lease replace the word “Its” as its

Figure 6 E: Please check the words “pB-Cas9-Neo”  and “pB-Cas9-neo”

Line 283: Grace medium”  please check the spellings.

Line 286: Then We extracted” the sentence may be replaced with “Then we extracted”

Line 299,303: (Fig7.C above). (Fig7.C down). Please rewrite the figure numbers as this is confusing to the readers

Line310: figure 7B: what is this 700, 1000….Please add the word ”bp” where necessary

Line 312: “arrows under the letters F, R, R1 indicates” Please check if the letters are correctly written

Line 306-308: Please add the conclusion portion in the MS and also add these sentences precisely in this portion

 Discussion:

Line 341-344: Besides, the promoter of 341 HSP70 gene…. promoter similar to the same type of promoters found by Chen et al [31]. Please rephrase the sentence as this is very lengthy and confusing for the readers

Line 349: Please check if the reference [42] is relevant to the text. Replace the reference with the latest (if necessary)

Line 358: Please add the word ribonucleoproteins (RNP) if necessary. As RNP abbreviation seems confusing for the readers

Line 357-360: To ensure efficient functioning of the CRISPR/Cas9 ….. system can be applied to insect cell lines. “Please rephrase the sentence seems lengthy and difficult to read.

Line 362: Please check if the “different transfection efficiency” is correctly written in the sentence or rephrase it to avoid confusion

 Materials and Methods

Line 379: Please replaced the word “Cultured” with cultured

Line 380: Please recheck if this is Grace medium or Grace’s medium (please change accordingly which is necessary)

Line385: Rephrase the sentence “The way of transfection without serum as follows”

Line 387-390: Please check the English grammar of the sentences, if necessary please rephrase accordingly

Line 393-395: The transfection method with serum …. transfection reagent and plasmid for 60h. Please rephrase the sentence

Line 396-397: incomplete sentence “In experiments in which pB-Cas9-Neo plasmid was co-transfected with pU6-shRNA 396 plasmid and pU6-sgRNA plasmid” please rephrase the sentence

Line 397-399: Rephrase the sentence “we distributed the pB-Cas9-Neo plasmid ….. dosage of the pU6 plasmid is 300ng/well.

Line 464: delete the word from the sentence “purchased from”

In methodologies, some subheadings are in italics and some are without italics. Please keep the consistency according to the journal’s style/author’s instructions

Line487: delete the word “and” by adding (,)

Please recheck the spellings in the titles of the tables S7 and S8

DAPI is the abbreviation. Please write the name of the abbreviation where firstly used in the text

References:

Please add suitable references for PCRs, RT-qPCR s, data analysis softwaresif used already designed primers, statistical data analysis

Reference writing is not with consistent style as some are in abbreviations and some are in full words, For example, Methods in enzymology. Please follow the journal’s style formatting/author's instructions for writing in all references Bridgman, P. C.; Brown, M. E.; Balan, I., Biolistic transfection. Methods in cell biology 2003, 71, 353-68.

Potter, H., Transfection by electroporation. Current protocols in immunology 2001, Chapter 10, Unit 10.15. Please check if the publisher’s name is included in these references. Please also check in all references of the manuscript

Round 2

Reviewer 2 Report

Article number 1912474 submitted to the International Journal of Molecular Sciences, IJMS, MDPI, entitled “Optimization and application of CRISPR/Cas9 genome editing in a cosmopolitan pest, diamondback moth” may be recommended for publication in International Journal of Molecular Sciences, IJMS, MDPI. Some suggested changes for example are in the comments portion.

Figure 6: Please mention the “bp” in the figure (10000, 7000, 4000, 2000, 1000, 700)

Please check the abbreviated forms of the journals name in the reference portion as some are in abbreviation and others are in full name “Methods in cell biology” Bioresource Technology, The Journal of cell biology, Pharmaceutical Research, Nucleic acids research, etc
